# Identification and Structure of Epitopes on Cashew Allergens Ana o 2 and Ana o 3 Using Phage Display

**DOI:** 10.3390/molecules28041880

**Published:** 2023-02-16

**Authors:** Ailin Zhang, HuiJuan Zhao, Shaohan Pei, Yue Chi, Xiuhua Fan, Jianqing Liu

**Affiliations:** 1College of Food Science and Biotechnology, Tianjin Agricultural University, Tianjin 300384, China; 2Baotou Teachers College, Inner Mongolia University of Science and Technology, Baotou 014030, China

**Keywords:** cashew nuts, allergens, phage display technology, epitope prediction

## Abstract

Background: Cashew (*Anacardium occidentale* L.) is a commercially important plant. Cashew nuts are a popular food source that belong to the tree nut family. Tree nuts are one of the eight major food allergens identified by the Food and Drug Administration in the USA. Allergies to cashew nuts cause severe and systemic immune reactions. Tree nut allergies are frequently fatal and are becoming more common. Aim: We aimed to identify the key allergenic epitopes of cashew nut proteins by correlating the phage display epitope prediction results with bioinformatics analysis. Design: We predicted and experimentally confirmed cashew nut allergen antigenic peptides, which we named Ana o 2 (cupin superfamily) and Ana o 3 (prolamin superfamily). The Ana o 2 and Ana o 3 epitopes were predicted using DNAstar and PyMoL (incorporated in the Swiss-model package). The predicted weak and strong epitopes were synthesized as peptides. The related phage library was built. The peptides were also tested using phage display technology. The expressed antigens were tested and confirmed using microtiter plates coated with pooled human sera from patients with cashew nut allergies or healthy controls. Results: The Ana o 2 epitopes were represented by four linear peptides, with the epitopes corresponding to amino acids 108–111, 113–119, 181–186, and 218–224. Furthermore, the identified Ana o 3 epitopes corresponding to amino acids 10–24, 13–27, 39–49, 66–70, 101–106, 107–114, and 115–122 were also screened out and chosen as the key allergenic epitopes. Discussion: The Ana o 3 epitopes accounted for more than 40% of the total amino acid sequence of the protein; thus, Ana o 3 is potentially more allergenic than Ana o 2. Conclusions: The bioinformatic epitope prediction produced subpar results in this study. Furthermore, the phage display method was extremely effective in identifying the allergenic epitopes of cashew nut proteins. The key allergenic epitopes were chosen, providing important information for the study of cashew nut allergens.

## 1. Popular Scientific Summary

Bioinformatic analyses are frequently used to predict protein properties related to antigenicity. We used two antigenic proteins present in cashew nuts to compare the results of the bioinformatics-based antigenicity predictions with those of the experimental epitope mapping. Several amino acid oligopeptides, representing the predicted and unpredicted epitopes, were expressed using phage display and screened with the pooled human sera of patients with a cashew nut allergy. The findings revealed that the computer predictions did not correlate very well with the presence of linear epitopes.

## 2. Introduction

The cashew plant (*Anacardium occidentale* L.) is a member of the Anacardiaceae family, which includes nine Anacardium species [1]. Cashew nuts are popular as snacks around the world, and their consumption has increased in recent years. However, some people are allergic to cashew nuts, so eating them poses a significant health risk to this population. With the increasing awareness of food allergies, a cashew nut allergy has become an important public safety and health problem [2]. Food allergy is a global public health problem which has increased significantly in the past decade, affecting the quality of life of consumers and increasing the demand for health service resources [3]. Food allergies affect approximately 2–4% of the population, peaking at 6–8% at the age of one year and gradually declining to 2–4% in older children and adults [4].

Every antigen has its own antigenic determinant or epitope. Epitopes are kinds of specific locations on antigens that could be recognized by immune cells [5]. An epitope typically consists of several amino acid (aa) residues, often (but not always) aligned along the peptide chain, which comes into physical contact with the lymphocytes during an immune reaction. Antibodies can specifically bind with epitopes. Because the binding sites between antibodies and antigens are relatively small, usually, the three-dimensional size of an epitope is only approximately 3 nm × 1.5 nm × 0.7 nm; the smaller size of the epitope typically allows about 5 to 7 aa or monosaccharide residues, and the epitope seldom consists of more than 20 aa [6,7].

Until now, several methods have been employed to study the epitopes in macromolecules. Bioinformatic analysis and the use of specific databases allow for a rapid and easy epitope prediction. For instance, Shan et al. successfully predicted the epitope segment of the β-soya globulin α subunit by using three different bioinformatics tools, DNAStar, the BepiPred 1.0, and the SOPMA web server [8]. Hu et al. predicted the dominant epitope in the peanut allergen by using SOPMA software. In contrast, the T-cell epitopes in peanut allergen Ara h 1, together with walnut allergen Jug r 2 and cashew allergen Ana o 1, were identified using various bioinformatics tools [9,10,11,12]. However, these methods are not accurate enough to predict such epitopes, and they need to be verified using in vitro assays, such as the highly powerful phage display method.

Phage display, an experimental method to screen peptides with certain properties, was first established by Smith [13] and subsequently used by others for various purposes [14,15]. Because of its continuous development, the phage display is currently being applied to several areas, including an antigen epitope study [16,17], molecular interaction research [18], vaccine development [19,20,21,22,23,24], and disease diagnosis and therapy [25,26,27,28].

Food allergy is a public health issue that has grown significantly in the last decade, affecting consumers’ quality of life and placing additional strain on healthcare resources. Furthermore, nut allergies can be fatal. The most common tree nut allergies in the USA are cashew nut allergy and walnut allergy, both of which are increasing globally [29]. So far, it has been determined that the condition is primarily caused by three proteins: Ana o 1 [30], Ana o 2 [31], and Ana o 3 [32]. Ana o 1 (a vicilin) and Ana o 2 (a legumin) are both members of the cupin superfamily, whereas Ana o 3 belongs to the prolamin superfamily [33,34]. Cashew has a low Ana o 1 content and a low sensitivity, making it difficult to isolate and obtain [35].

Here, we used bioinformatics tools to predict the epitopes on the Ana o 2 and Ana o 3 proteins and used the phage display method to verify these predictions by epitope identification. Phage libraries were screened using the human sera of patients with a cashew allergy. Our study provides the theoretical foundation for a better characterization of allergens, thereby assisting the development of hypoallergenic foods in the future.

## 3. Results

### 3.1. Identification of Cashew Allergens

The allergens in cashews were isolated using Sephadex G-150 column chromatography, followed by SDS-PAGE electrophoresis and Western blot analysis. The Sephadex purification yielded only four proteins in relatively large quantities. Moreover, the IgE Western blotting of the allergens also revealed two major bands, approximately 33 KDa and 17 KDa. The results of this experiment confirmed the previously reported findings of purified Ana o 2 and Ana o 3 being the two main allergens of cashew nuts. We further performed a structural analysis of these purified proteins.

The main advantage of mass spectrometry in the detection of food allergens is the sequence determination of the peptides and proteins by the protein spectrum. Specifically, a protein was cut into small fragments by specific enzymatic or chemical hydrolysis, and the molecular weight of each product was detected using mass spectrometry. The obtained data were applied to the database to obtain the corresponding known protein and to determine the sequence of the protein to be measured, as shown in Figure 1A. The impact of detection during processing can thus be effectively avoided. The Ana o 2 and Ana o 3 allergens were also identified. The mass spectrometric profile data were compared to the UniProt-Anacardium Occidentale (downloaded on 20 April 2019) and NCBI-Anacardium Occidentale databases. A total of 60,842 profiles and 5 proteins were obtained. The number of peptides obtained from UniProt-*Anacardium occidentale* was 147, and from the NCBI-*Anacardium occidentale*, it was 148. The measured fragment sequences were compared with those obtained from the NCBI database, and two of them were identified to match the sequences of Ana o 2 and Ana o 3. The sequences of Ana o 2 and Ana o 3 are as follows:

The CD was used to determine the structure of the protein solution, and the CD spectrum in the range of 180–260 nm was provided in the protein secondary structure. From Figure 1B, we found that between 180 and 195 nm, the changes in the α–helix, β–turn, and random coil were relatively stable. The α–helix decreased from 30.70% to 12.20%, the β–turn increased from 17.70% to 22.40%, and the random coil increased from 32.80% to 54.00%, with an overall trend of a decreasing α–helix and an increasing β–turn and random coil. Because of their bulge structures, the β–turn and random coil easily bind to antibodies to form epitopes, whereas the α–helix is not easily deformed and does not easily bind to antibodies to form epitopes, such as tropomyosin, a coiled–coil protein with an α–helix structure that becomes a major allergen from shellfish [36]. Furthermore, a high frequency of β–turn and random coil occurrence was found in the entire amino acid sequence of Ana o 2 and Ana o 3, indicating a high possibility of epitope formation in this region.

### 3.2. Prediction of the Allergen Secondary Structure with Bioinformatic Analysis

The secondary structures of the allergens Ana o 2 and Ana o 3 were predicted using Protean, a component of the DNAStar software, in conjunction with Swiss-model and PyMoL software; the results are shown in Figure 2 (bioinformatic analysis of Ana o 2 (A) and Ana o 3 (B) features). Those located on the outside of the protein were chosen from among all the identified linear epitopes. Furthermore, those found in the flexible regions and rich in beta–sheets and flexible loops were chosen. Any predicted protruding structure would allow antibodies easy access to the epitope. Therefore, domains that met these requirements were considered potential epitope candidates with a high probability.

The hydrophilicity, surface accessibility, and flexibility of the protein domains are important parameters affecting the antigenic properties. A hydrophilicity above 0, an antigen index above 0, and a surface accessibility of above 1 would, in combination, increase the chance of a protein region representing an epitope. Therefore, a protein domain would have a high surface accessibility as well as the flexibility to fold easily, increasing the exposure at the surface of a three-dimensional protein structure.

The antigen index predicts the likelihood of a protein containing epitopes. As shown in Figure 2B, at least six distinct regions and a large continuous portion of Ana o 2 and Ana o 3 had a high antigen index. To put such predictive indices to the test, we created epitopes with varying lengths and scores based on the antigenicity, hydrophilicity, and surface probability, as shown in Figure 2. Ana o 2’s peptides were shorter than those of Ana o 3. As the controls for Ana o 3, two overlapping peptides (Ana o 3–1 and Ana o 3–2) located in regions with negative hydrophilicity and antigenic index values were used. The amino acid sequences were converted to nucleotide sequences through the EMBOSS Backtranseq software (https://www.ebi.ac.uk/Tools/st/emboss_backtranseq/, accessed on 23 Jun 2021); the results are summarized in Table 1.

### 3.3. Cloning of the Epitope Libraries

Oligonucleotides representing the predicted epitope sequences were cloned into the plasmid pCANTAB5E and transformed into the *E. coli* host DH5alpha. After the insert length was confirmed (Figure 3), the plasmids were electroporated into TGI cells. Three colonies were selected from each transformation plate and sent out for sequencing. After the confirmation of the correct inserts, they were used for phage display screening.

### 3.4. Phage Display Screening

Human serum derived from patients with a cashew nut allergy was used to screen the phage display library. This resulted in the identification of major epitopes whose OD_450_ readings are listed in Table 2. Positive signals were obtained for the peptides Ana o 2–3, Ana o 2–4, and Ana o 2–6, as well as a weak positive signal for Ana o 2–9. Despite having a high antigenicity, hydrophilicity, and surface probability, as well as the predicted coil and turn structure, the peptide Ana o 2–2 was not recognized by the pooled antisera. Apart from the presence of a turn but no coil motive, Ana o 2–3 exhibited good but not maximum scores for hydrophilicity, antigenicity, and surface exposure, and it was antigenic. The strongest signal was produced by Ana o 2–4, a peptide of seven amino acids (Table 1), which is located within a stretch with high hydrophilicity (though not producing top scores) and produced maximum scores of antigenicity, combined with a predicted turn and coil region (Figure 3A). Although Ana o 2–5 also contained predicted coil and turn regions together with high predicted hydrophilicity and maximum antigenicity scores, it did not produce positive results in an ELISA. Ana o 2–6 combined maximum antigenicity scores with the presence of a turn but not a coil region; its hydrophilicity was relatively low, and the surface exposure index was even lower. Nevertheless, Ana o 2–6 was recognized as an epitope by the pooled human serum. The weak but positive ELISA results from Ana o 2–9 were unexpected considering the obtained prediction scores.

The Ana o 3 peptides produced weaker ELISA signals than the Ana o 2 peptides. Ana o 3–10 produced the strongest ELISA signal, despite having lower hydrophilicity and surface probability scores than Ana o 3–3, which did not represent an epitope. The two overlapping peptides, Ana o 3–1 and Ana o 3–2, were detected by the human antiserum and scored negative for the hydrophilicity, antigenicity, and surface probability (Figure 3B, Table 2). Therefore, these findings indicated that these bioinformatics tools performed poorly in predicting epitopes, at least for these two proteins.

## 4. Discussion

We compared the Ana o 2 and Ana o 3 epitopes predicted by in silico analysis to the epitopes identified by phage display technology. The study’s goal was to examine the allergen antigenic peptides. It is important to find IgE antibody samples and purify the allergen−specific IgE for the phage. In our experiment, we used an indirect ELISA with a mouse anti−human IgE monoclonal antibody−coated to capture human IgE in pooled samples, avoiding cross−reactivity with other pooled samples or binding with epitopes. We also detected and discovered that the sIgE value (3.2) of the pooled samples was much greater than the 0.35 KUA/L in the patient group and less than 0.35 KUA/L in the healthy control group (0.08). All of this resulted in the important epitopes for the allergic reaction that we are studying being IgE−binding epitopes rather than others−binding epitopes.

Phage display technology was used in this study to determine the main allergen epitopes, primarily because the traditional enzyme−linked immunosorbent assay would inactivate all of the proteins and viruses, resulting in more nonspecific binding reactions of antigen epitopes and interfering with the experimental results. The phage display technology has a higher activity and less nonspecific binding. Because the synthetic peptide’s molecular weight was less than 100 bp, many PCR experiments failed, and the single-strand hybridization complementary method was finally chosen to form double strands [37]. When building the transformation library, the electric transformation method was used instead of chemical transformation because it was more efficient. During the experiment, it was discovered that if the peptide was not purified, it could not connect with the bacteriophage. The analysis could be attributed to the dimer-induced inhibition of its binding. Therefore, gel cutting and purification were performed prior to the transformation in order to remove the impurities, such as dimers from the product and improve the linkage rate [38].

From the study of the bioinformatic analysis of the Ana o 2 (A) and Ana o 3 (B) features, between 180 and 195 nm, the changes in the α−helix, β−turn, and random coil were relatively stable. The α−helix decreased from 30.70% to 12.20%, the β−turn increased from 17.70% to 22.40%, and the random coil increased from 32.80% to 54.00%, with an overall trend of a decreasing α−helix and an increasing β−turn and random coil. Because of their bulge structures, the β−turn and random coil easily bind to antibodies to form epitopes, whereas the α−helix is not easily deformed and does not easily bind to antibodies to form epitopes, such as Tropomyosin, a coiled−coil protein with an α−helix structure that becomes a major allergen from shellfish. Furthermore, a high frequency of β−turn and random coil occurrence was found in the entire amino acid sequence of Ana o 2 and Ana o 3, indicating a high possibility of epitope formation in this region.

A previous study used immunoblotting and 15 aa long peptides [39] to perform a systematic epitope screening on Ana o 2. The study identified seven peptides that bound to human IgE, of which aa 105–119 contained Ana o 2–3. The peptide with aa 113–127, on the other hand, did not produce positive results; it contained Ana o 2–4, which we classified as an epitope. The systematic immunoblot screening performed by Wang et al. did not completely cover our epitope Ana o 2–6, while Ana o 2–9 was missed by the corresponding peptide with aa 217–231. Ana o 3 was subjected to a similar systematic epitope screening, with aa 57–68, 72–83, and 102–113 identified as strongly reactive. Our analysis did not confirm the first two, while the third overlapped with our Ana o 3–8 and Ana o 3–9. Immunoblotting can only detect linear epitopes that are not destroyed by denaturation [40]. Phage display screening, on the other hand, allows for some structural features to be conserved, which improves the sensitivity. The use of pooled sera rather than individual serum may have diluted out some of the minor epitopes. Furthermore, the validation of the predicted epitopes is highly dependent on the patient serum; however, the volume of 10 people’s mixed serum samples was sufficient to compensate for this.

Therefore, the fraction of aa producing allergen epitopes was 5.5%. Ana o 3 had a much larger fraction because 56 aa produced at least six and possibly seven epitopes, whereas the entire protein was only 138 aa long, resulting in an epitope−producing aa fraction of 40.6%. Ana o 3 is thought to be more likely to cause an allergic reaction than Ana o 2 due to the higher number of identified epitopes and a much smaller size.

## 5. Materials and Methods

### 5.1. Preparation of Protein Samples and Patients’ Samples

The protein was resolved using Omni PAGE precast gel (4–15%). The protein samples were prepared by mixing with 5X loading buffer (4:1 by volume) and heating for 5 min, followed by cooling at room temperature. The cashew allergen protein was purified using glucan G−150 column chromatography [41]. SDS−PAGE [42] and Western blotting were performed as described in a previous study. Subsequently, 10 µL of the prepared samples were loaded onto the gel (a maximum of 60 µL per well). The gel was run at 150 V until the bromophenol blue dye front reached the gel bottom (40–50 min). Half of the gel was then incubated in a staining solution for 30 min with gentle shaking. For destaining, the gel was incubated with a destaining solution and photographed using a gel imaging system. The other half of the gel was used for Western blotting, and the proteins were visualized by immune detection.

After people with a cashew allergy eat cashews, a special substance (human IgE antibody) appears in their blood, causing hypersensitivity [43]. This study followed all clinical study procedures, as described in the introduction [44]. So, we created two groups: one for patients and one for healthy controls. The patients with cashew nut allergies were identified as the patient group; all of them had a specific IgE antibody value of cashew no less than 0.35 KUA/L with typical and compound clinical allergy symptoms, such as anaphylactic symptoms on the skin or in the respiratory or gastrointestinal areas detected by doctors, and the sera from 10 cases with a cashew nut allergy as the positive serum samples were provided by our collaborators at the Xinzhou People’s Hospital. The healthy control group consisted of healthy participants with no obvious history of an allergy, as well as those from Xinzhou People’s Hospital. Both groups of participants aged 18–59 years (regardless of gender) were free of other diseases. We used a diagnostic kit to confirm that the patients and control groups did not have any other food allergies when we collected the samples for our group (hob–biotech, Suzhou). The information from these samples is shown in Table 3, which includes sex, age, cashew–specific IgE, cashew reaction history, and diagnosis method. If the SIgE value of the cashew was not less than 0.35 KUA/L, with anaphylactic symptoms on skin or in the respiratory and gastrointestinal areas diagnosed by doctors, it was diagnosed to be that of a cashew allergy patient.

### 5.2. Mass Spectrometry Sequencing

Protein samples must be pretreated before the enzymatic hydrolysis of protein solutions can take place. To begin, we took some protein samples and mixed them with 20 µL of 2 M urea. After 30 min, 20 µL of 10 mM dithiothreitol (DTT) (dissolved in 25 mM NH_4_HCO_3_) were added and incubated at 37 °C in a water bath for 1 h. After that, it was filled with 20 µL of 55 mM indole–3–acetic acid (IAA) (dissolved in 25 mM NH_4_HCO_3_) and left in the dark for 30 min. The samples were then fully mixed and incubated overnight at 37 °C with 0.2 µg of trypsin. The samples were centrifuged the next day and then desalted using the C18 Zip Tip column. The protein solutions were then centrifuged, vacuum freeze–dried, and kept at −20 °C.

The mass spectrometry analysis was run. The peptide fragment was resolubilized in 20 µL of 2% methanol and 0.1% formic acid. After centrifuging the mixture at 12,000× *g* rpm for 10 min, the supernatant was aspirated for sample loading. Following that, 10 µL of the sample was loaded for 15 min using the sandwich method, with a loading pump flow rate of 350 nL/min and a separation flow rate of 300 nL/min.

Following the mass spectrometry sequencing, the peptide sequencing data were compared to those found in the UniProt-*Anacardium occidentale* database (downloaded on 20 April 2019) and the NCBI–*Anacardium occidentale* database. Maxquant (an information retrieval package) was also used to analyze the proteomic data.

### 5.3. Determination of Secondary Structures by Using Circular Dichroism (CD)

First, the background was measured without any objects being placed in the sample pool. Following that, the sample solution was injected into the reader’s cuvette (the machine: Circular Dichroism Spec, Applied PhotoPhysics, APL). Meanwhile, the samples were placed in a cuvette and detected three times at a 0.5 nm bandwidth, 1.0 nm step, 180–260 nm wavelength, and 0.5 s time-per-point.

### 5.4. Bioinformatics Analysis

The allergen sequences obtained by the mass spectrometry sequencing were compared with those obtained from the NCBI database (https://www.ncbi.nlm.nih.gov/protein/AAN76862.1?report=genpept, accessed on 8 April 2021). Then, the secondary structures of the predicted allergenic proteins were compared using the Protean program in DNAstar, with the structure and binding of the proteins determined using the CD.

### 5.5. Epitope Prediction

The aa sequences of allergens Ana o 2 and Ana o 3 were extracted from the NCBI database (accession number AF453947_1 and AAL91665, respectively). The antigenic epitopes of these two proteins were predicted using the default settings of the DNAstar and PyMoL software, both of which are part of the Swiss model (https://swissmodel.expasy.org/interactive, accessed on 8 April 2021), along with predictions of their hydrophilicity, flexibility, and surface accessibility. Jin Wei Zhi Biological Co., Ltd. synthesized the DNA oligonucleotides of the predicted epitopes, with upstream and downstream primers designed to contain the *Xma*I and *Kpn*I restriction sites (forward primer: CAGCCCCGGTACCCCG, reverse primer: GCGGCCGTCCCCGGG), which added 33 bp to the length of each peptide nucleotide sequence [45].

### 5.6. Epitope Cloning and Expression

The pCANTAB5E phage display system was used to express the predicted epitopes. To see if the peptides were correctly cloned into the plasmid, the epitopes were cloned and expressed in *E. coli* DH5a cells. The pCANTAB5E plasmid was transformed into *E. coli* DH5α, cultured with ampicillin as the selective antibiotic, and extracted using an alkaline lysis plasmid extraction kit for this purpose (Omega). The plasmid DNA was digested with the restriction enzymes *Kpn*I and *Xma*I (NEB), separated on a 1.5% agarose gel, and purified using a gel recovery kit (Promega). The synthesized primers were used separately to amplify the entire predicted peptide in a reaction volume of 10 µL using Taq polymerase. The PCR conditions underwent pre-denaturation for 1 min and 20 cycles of 1 min at 95 °C, 30 s at 55 °C, and 30 s at 72 °C, followed by a 4 min terminal extension at 72 °C. After two cycles, the two complementary primers were combined and annealed for 1 h at 37 °C. The amplicons were separated on 5% agarose gels before being cut out, purified, and ligated into the pCANTAB5E plasmid DNA. The ligation mixture was transformed into competent *E. coli* DH5α cells, and the DNA extracted from single colonies was sequenced by a third party (Sangon Biotech) [46].

The recombinant plasmids were then isolated, quantified, and electroporated into TGI competent cells; the transformed cells were diluted with super optimal broth with a catabolite repression (SOC) culture medium and cultured overnight at 37 °C in 2× YT/Amp/Glu plates with 2% glucose and 100 µg/mL. The colonies on the plates were sequenced the following day.

### 5.7. Phage Peptide Expression in Microtiter Plates

A bacterial suspension (50 µL) of each primary library was added to a 96-well plate together with a 200 µL 2XYT/AMP/Glu culture medium and incubated at 37 °C overnight with a 250 rpm rotation. The cultures were stored at −70 °C with an equal volume of 100% glycerol. The subcultures were grown in 96-well plates until an OD_600_ of 0.4–0.6 was obtained. To these exponentially growing cells, the helper phage M13KO7 (NEB) was added at a ratio of 10 or 20:1 (phage to bacteria) and incubated at 37 °C for 30 min. Following centrifugation, the cell pellets were resuspended in 150 µL of 2XTY/Amp/Kan. Because the expression of the recombinant peptides depends on the osmotic expression of the *lacZ* promoter with glucose removed in this step, the bacteria were cultured overnight at 37 °C and 300 rpm. Following centrifugation, 50 µL of the supernatant obtained from each well were used for the phage enzyme-linked immunosorbent assay (ELISA).

### 5.8. Identification with Phage ELISA Method

The plate was coated with a mouse anti-human IgE monoclonal antibody (purchased from Jackson immunoResearch Laboratories Inc.) and incubated overnight at 4 °C. The coated test plates were washed three times with PBST the next day. These plates and an uncoated control plate were blocked for 45 min in PBS with 2% BSA. After washing, the pooled human sera derived from ten individuals with a cashew allergy and the pooled human sera from the control group were added into the wells (100 µL/well) and incubated for 1 h (PBST). After washing the plates, each phage supernatant was used as the antigen (mixed 1:1 with 1% MPBS) and added into the wells. Finally, a 1:4000 dilution of monoclonal mouse anti-M13-HRP (MBL) was used as the secondary antibody. After the final wash, the substrate solutions were added to the plates for 15 min of incubation. After adding the stop solution, the plates were read on a Gen5 plate reader (BioTek Co., Ltd., Santa Clara, CA, USA).

## 6. Conclusions

Epitopes were identified on two proteins derived from cashew nuts by using phage library expression and screening with human sera obtained from patients with a cashew nut allergy. The bioinformatic predictions correlated poorly with the experimentally verified epitopes. Of the identified epitopes, Ana o 3 is more likely to cause allergic reactions than Ana o 2. Therefore, further research should focus on changing the composition or structure of the identified protein domains to produce low-allergen nuts. We conclude that cashew nut allergen epitopes can be expressed, and their physicochemical properties can be studied, by using the bacteriophage display technology, which provides a theoretical basis for allergen studies in the future.

## Figures and Tables

**Figure 1 molecules-28-01880-f001:**
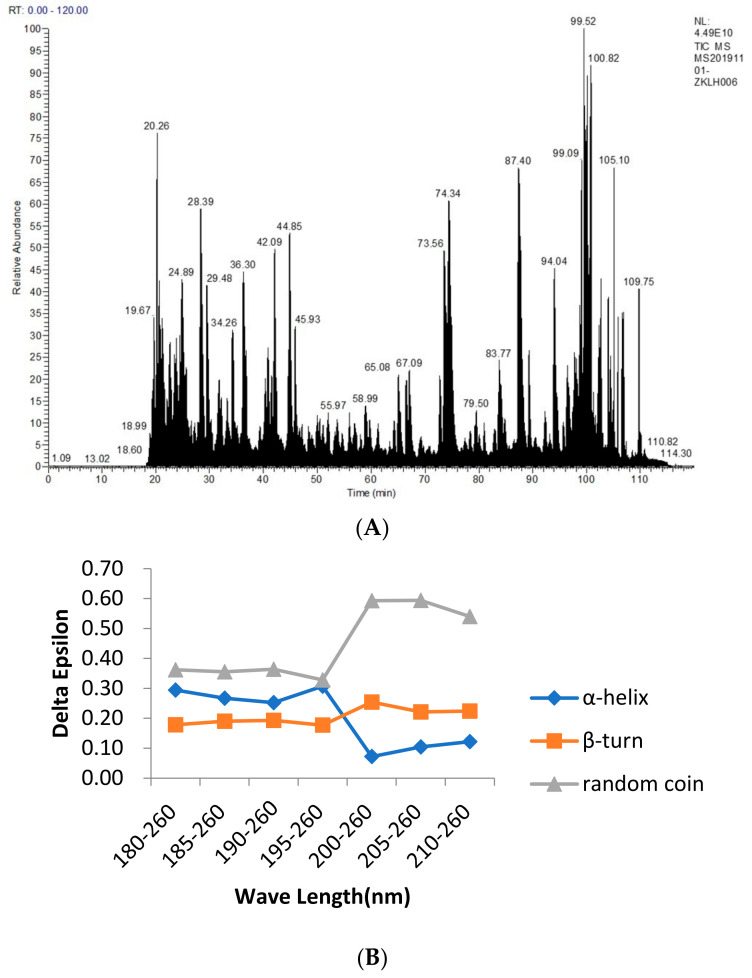
Mass spectrometry analysis of the cashew allergenic proteins (**A**) and the circular dichrogram (**B**) to identify cashew allergenic proteins. Figure (**A**) is from the results of the protein sequencing analysis of cashew allergens. The main purpose of this experiment is to identify two major allergens by Western blotting and obtain the sequences of the two allergens by mass spectrometry sequencing, as it is measured by liquid mass spectrometry. ESI mass spectrometry provides the total ion map. After the NCBI database query, the allergen sequence and the results obtained by mass spectrometry are compared. The secondary structure predicted by the Protein program in DNAstar is compared with the protein structure determined by circular dichroism.

**Figure 2 molecules-28-01880-f002:**
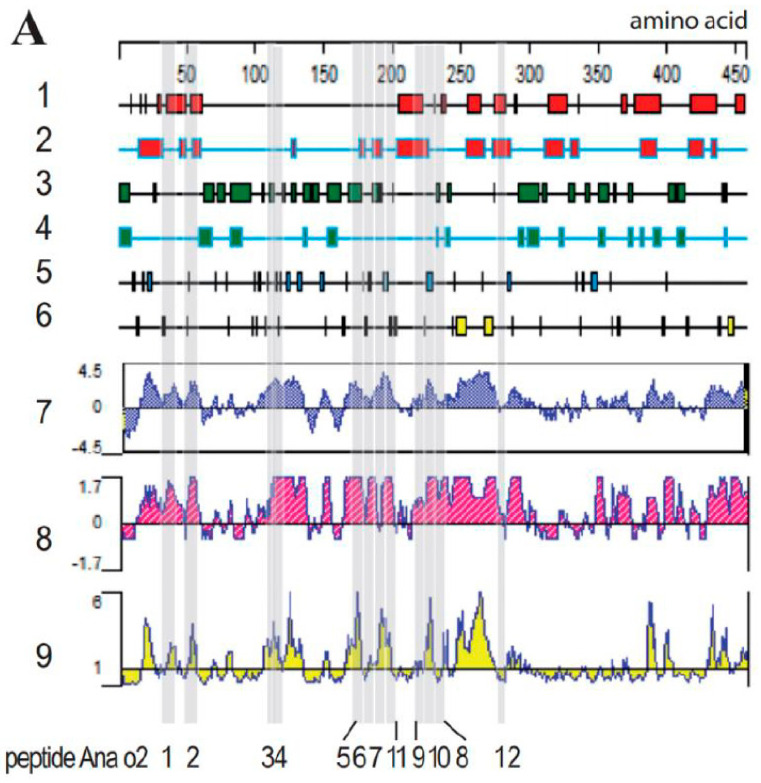
Bioinformatic analysis of the Ana o 2 (**A**) and Ana o 3 (**B**) features. Lanes 1–6: secondary structure analysis revealing alpha regions by using the Garnier–Robson (1) and Chou–Fasman methods (2), beta regions by using the Garnier–Robson (3) and Chou–Fasman methods (4), turn regions by using the Garnier–Robson method (5), and coil regions by using the Garnier–Robson method (6). Further lanes reveal hydrophilicity by using the Kyte–Doolittle method (7), the antigenic index by using the Jameson–Wolf method (8), and the surface probability by using the Emini method (9). Grey shading in both panels corresponds to the produced peptides, whose names are listed below the panels.

**Figure 3 molecules-28-01880-f003:**
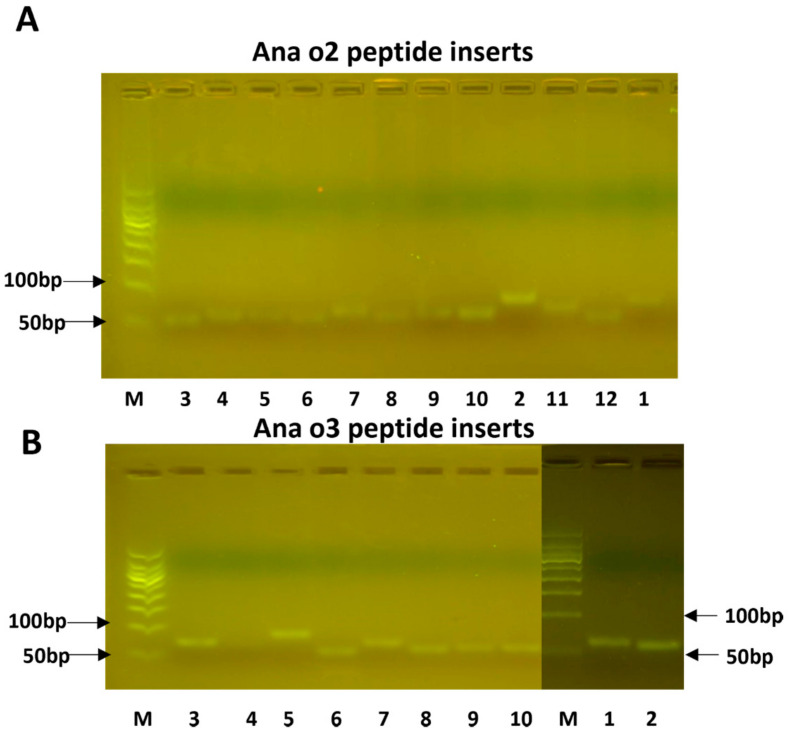
The cloned peptide inserts for Ana o 2 (**A**) and Ana o 3 (**B**). The numbers below the lanes correspond to the peptide numbers provided in Table 1. Lane M represents a size marker.

**Table 1 molecules-28-01880-t001:** Peptide sequences of the predicted epitopes and the corresponding nucleotide sequences.

Name	Amino Acid Position	Amino Acid Sequence	Nucleotide Sequence
Ana o 2–1	31–41	DALEPDNRVEY	GACGCCCTGGAGCCCGACAACAGGGTGGAGTAC
Ana o 2–2	47–57	EAWDPNHEQFR	GAGGCCTGGGACCCCAACCACGAGCAGTTCAGG
Ana o 2–3	109–113	QQGRQ	CAGCAGGGCAGGCAG
Ana o 2–4	114–120	QGQSGRF	CAGGGCCAGAGCGGCAGGTTC
Ana o 2–5	171–182	FHLAGNP	TTCCACCTGGCCGGCAACCCC
Ana o 2–6	182–187	PKDVFQ	CCCAAGGACGTGTTCCAG
Ana o 2–7	188–194	QQQQHQS	CAGCAGCAGCAGCACCAGAGC
Ana o 2–11	195–203	RGRNLFSGF	AGGGGCAGGAACCTGTTCAGCGGCTTC
Ana o 2–9	219–225	IKQLKSE	ATCAAGCAGCTGAAGAGCGAG
Ana o 2–10	226–232	DNRGGIV	GACAACAGGGGCGGCATCGTG
Ana o 2–8	233–239	KVKDDEL	AAGGTGAAGGACGACGAGCTG
Ana o 2–12	284–288	ENTND	GAGAACACCAACGAC
Ana o 3–1	10−24	AFAVLLLVANASIYR	GCCTTCGCCGTGCTGCTGCT GGTGCCAACGCCAGCATCTACAGG
Ana o 3–2	13−27	VLLLVANASIYRAIV	GTGCTGCTGCTGGTGGCCAA CGCCAGCATCTACAGGGCCATCGTG
Ana o 3–3	30–39	EEDSGREQSC	GAGGAGGACAGCGGCAGGGAGCAGAGCTGC
Ana o 3–4	39–48	QRQFEEQQR	CAGAGGCAGTTCGAGGAGCAGCAGAGG
Ana o 3–6	66–71	YNQRQE	TACAACCAGAGGCAGGAG
Ana o 3–7	83–92	VDRRCRCQNL	GTGGACAGGAGGTGCAGGTGCCAGAACCTG
Ana o 3–8	101–107	QQEQIKG	CAGCAGGAGCAGATCAAGGGC
Ana o 3–9	108–115	EEVRELYE	GAGGAGGTGAGGGAGCTGTACGAG
Ana o 3–10	116–123	TASELPRI	ACCGCCAGCGAGCTGCCCAGGATC
Ana o 3–5	124–138	CSISPSQGCQFQSSY	TGCAGCATCAGCCCCAGCCA GGGCTGCCAGTTCCAGAGCAGCTAC

**Table 2 molecules-28-01880-t002:** Phage-ELISA results and the predicted antigenicity, hydrophilicity, and surface probability of the peptides.

Peptide	^Δ^ELISA Value	Hydrophilicity Index ^1^	Antigenicity Index ^2^	Surface Probability ^3^
Ana o 2–1	−0.042	2.0	1.7	2.0
Ana o 2–2	−0.023	2.8	1.7	3.5
**Ana o 2–3**	**1.558**	2.0	0.8	2.3
**Ana o 2–4**	**3.532**	3.0	1.7	3.8
Ana o 2–5	−0.044	2.8	1.7	6.0
**Ana o 2–6**	**1.835**	1.0	1.7	1.0
Ana o 2–7	−0.023	3.8	0.8	4.8
Ana o 2–8	−0.018	0.5	1.6	−1.0
**Ana o 2–9**	**0.292**	1.2	0.6	0.8
Ana o 2–10	−0.031	2.8	1.7	5.8
Ana o 2–11	−0.105	2.0	1.7	2.8
Ana o 2–12	−0.024	0.4	0.4	−1.0
**Ana o 3–1**	**0.152**	−3.0	−0.6	−1.0
**Ana o 3–2**	**0.513**	−3.0	−0.6	−1.0
Ana o 3–3	−0.024	2.8	1.7	2.5
**Ana o 3–4**	**0.66**	3.2	1.7	4.0
Ana o 3–5	−0.106	1.1	0.6	1.0
**Ana o 3–6**	**0.216**	3.2	1.7	6.0
Ana o 3–7	−0.099	2.0	1.7	4.0
**Ana o 3–8**	**0.305**	2.3	1.1	3.5
**Ana o 3–9**	**0.227**	1.5	0.9	3.0
**Ana o 3–10**	**0.358**	1.3	0.9	2.2

^1^. Peak values correspond to lane 7 of Figure 3. ^2^. Peak values correspond to lane 8 of Figure 3. ^3^. Peak values correspond to lane 9 of Figure 3. ^Δ^ELISA values are calculated by the peptide sample OD value minus the OD value of the negative control with those above 0 written in bold.

**Table 3 molecules-28-01880-t003:** The patients’ hospital records of cashew allergy.

Group Number	Sample ID	Gender	Age	Specific IgE of Cashew *	Total IgE	Reaction History and Symptoms of Cashew Occurred	Diagnosis Method
Patients Group	SA1	Male	26	0.59	<100	Skin itch, erythema, rhinitis	>0.35 positive
	SA2	Male	22	0.69	<100	Skin itch, erythema, rhinitis	>0.35 positive
	SA3	Female	43	0.47	<100	Skin itch, erythema, rhinitis	>0.35 positive
	SA4	Male	27	0.43	<100	Respiratory tract sneezing, erythema, rhinitis	>0.35 positive
	SA5	Male	35	2.1	<100	Emesis, erythema, rhinitis, skin itch	>0.35 positive
	SA6	Female	19	1.6	<100	Erythema, rhinitis, skin itch	>0.35 positive
	SA7	Female	36	0.39	<100	Skin itch, erythema, rhinitis	>0.35 positive
	SA8	Male	54	0.57	<100	Skin itch, erythema, rhinitis	>0.35 positive
	SA9	Female	42	0.73	<100	Rhinorrhea, erythema, rhinitis, skin itch	>0.35 positive
	SA10	Male	33	7.8	>200	Urticaria, skin itch, erythema, rhinitis	>0.35 positive
Healthy Control	HC1	Female	23	0.09	<20	Not found	>0.35 positive
	HC2	Female	18	0.13	<20	Not found	>0.35 positive
	HC3	Female	27	0.12	<20	Not found	<0.35 negative
	HC4	Male	34	0.09	<20	Not found	<0.35 negative
	HC5	Male	36	0.05	<20	Not found	<0.35 negative
	HC6	Female	47	0.02	<20	Not found	<0.35 negative
	HC7	Female	42	0.03	<20	Not found	<0.35 negative
	HC8	Male	53	0.02	<20	Not found	<0.35 negative
	HC9	Female	58	0.03	<20	Not found	<0.35 negative
	HC10	Male	32	0.08	<20	Not found	<0.35 negative

* Here if SIgE value of cashew is not less than 0.35 KUA/L with allergic symptoms in skin, respiratory and gastrointestinal detected by doctors, it will diagnose to be cashew allergy patients.

## Data Availability

Not applicable.

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
