# Peer review of "Identification and Structure of Epitopes on Cashew Allergens Ana o 2 and Ana o 3 Using Phage Display"

_molecules, 2023, doi:10.3390/molecules28041880_

Round 1

Reviewer 1 Report (Previous Reviewer 1)

The scientific soundness of this manuscript depends in part on the utilization of serum from true allergic patients.  In the Supplemental table I provided, several items need to be clarified in the reaction history.  

1. In every patient asthma is recorded in the reaction history. If this is referring to respiratory symptoms within two hours after ingesting cashew then respiratory should be written in the table and not asthma.  If this is referring to the diagnosis of asthma in these patients, then it should be removed altogether from the table and is not part of the reaction history.

2. If the erythema eczema is a diagnosis it also should be removed from the reaction history. If within two hours after eating the cashew there was erythema, then just write erythema and not eczema. 

3. In SA6, diarrhea is not an Ige-mediated allergic response.

4. Did any of the other patients besides SA5, have abdominal pain?

5. The word anaphylaxis refers to reactions in which two or more systems are involved. Please check manuscript as to whether "allergic" symptoms ... should be used instead of anaphylaxis.

Author Response

Response to reviewer

Dear  reviewers:

Thank you so much for your professional, patient and kindly guidance to us once more time. Based on your professional comments, we have carefully discussed and revised the draft. And now reply them one by one ass follows. Please check the BLUE part and meanwhile we also have marked the RED highlight part in the revised text. Please guide us. Thanks a lot.

Q1: Dear Dr. Zhang, One of the reviewers has suggested that your manuscript should undergo extensive English revisions. Note that extensive English editing is not included in the APC. This may not accurately reflect the English level of your manuscript but we recommend that you further check the English during revision. If the reviewer does not provide detailed comments, you may ask the assistant editor to contact the reviewers for more specific comments. We propose that you have your manuscript checked by a native English-speaking colleague, or use a paid editing service. We also offer paid editing services at https://www.mdpi.com/authors/english. If you use an alternative service that provides a confirmation certificate, please send a copy to the Editorial Office. Best, English Editing Department [email protected]

Reply1: Thank you for your remind. Following your suggestion, we have sent our revised manuscript to English Editing Department and have polished with our certificate as follows:

Thank you.

Q2: 1. In every patient asthma is recorded in the reaction history. If this is referring to respiratory symptoms within two hours after ingesting cashew then respiratory should be written in the table and not asthma.  If this is referring to the diagnosis of asthma in these patients, then it should be removed altogether from the table and is not part of the reaction history.

Reply2: Thank you for your suggestion. We have modified the Supplementary Table 1. Please kindly check the Supplementary Table 1. Thanks.

Q3: 2. If the erythema eczema is a diagnosis it also should be removed from the reaction history. If within two hours after eating the cashew there was erythema, then just write erythema and not eczema. 

Reply3: Thank you for your suggestion. We have modified the Supplementary Table 1. And just write erythema. Please kindly check the Supplementary Table 1. Thanks.

Q4: 3. In SA6, diarrhea is not an Ige-mediated allergic response.

Reply4: Thank you. We have deleted diarrhea. Thanks a lot.

Q5: 4. Did any of the other patients besides SA5, have abdominal pain?

Reply5: Thank you. We have checked all our patients’ information again, and found that only SA5 has abdominal pain. Also we have searched some reference about this area and found that only few patients has abdominal pain situation of gastrointestinal symptoms. Thanks again.

Q6: 5. The word anaphylaxis refers to reactions in which two or more systems are involved. Please check manuscript as to whether "allergic" symptoms ... should be used instead of anaphylaxis.

Reply6: Thanks for your suggestion. We have replaced the anaphylaxis into allergic in the whole manuscript. Please kindly check the present manuscript. Thank you.

Thanks you so much for guiding us! We are really looking forward to your reply.

Have a GOOD day!

Yours sincerely

Ailing Zhang & Jianqing Liu

Feb 03rd, 2023

Reviewer 2 Report (Previous Reviewer 2)

The authors have addressed my previous concerns

Author Response

Response to reviewer

Dear  reviewer:

Thank you so much for your professional, patient and kindly guidance to us once more time.  Thanks a lot.

We also offer paid editing services at https://www.mdpi.com/authors/english. If you use an alternative service that provides a confirmation certificate, please send a copy to the Editorial Office. Best, English Editing Department [email protected]

Thanks you so much for guiding us! We are really looking forward to your reply.

Have a GOOD day!

Yours sincerely

Ailing Zhang & Jianqing Liu

Feb 03rd, 2023

Round 2

Reviewer 1 Report (Previous Reviewer 1)

The term for skin itch is pruritis. Please change to that term.

This manuscript is a resubmission of an earlier submission. The following is a list of the peer review reports and author responses from that submission.

Round 1

Reviewer 1 Report

Thank you for your revisions and the placement of the supplementary table.

Since the impact of your paper depends on the presumption of IgE-mediated allergic disease, some clarifications of the diagnostic method are necessary.

q1-What do the authors mean by skin itch in the supplementary table. Was this after the ingestion of cashew ?

q2-Did patients have more than one symptom ? The authors use the term anaphylactic on line 112 and yet only one symptom is given.

Reviewer 2 Report

Zhang and colleagues have worked to improve their manuscript.  Unfortunately, many deficiencies remain.

1.       The description of the subjects leaves this reviewer with concerns.  The cashew-specific IgE values are quite low and the clinical characteristics are quite non-specific.  This cannot be changed and limits the value of the manuscript.

2.       The description of the phage work is not consistent.  How does one use the “phage supernatant” (line 204) to identify phage that should be in the pellet?  At line 360, they say “pan the page” so it seems they cannot be looking at supernatants.

3.       In line 186, the authors say “Phage antibody expression” yet the manuscript is about peptides.

4.       Figure 1B does not show a clean peak.  Figure 2 demonstrates that there are multiple proteins in the material studied.  That said, the evaluation of peptides in the phage expression system and the ELISA used here do not need purified proteins.  So, there seems to be no value to showing Figures 1 and 2.

5.       Why are some ELISA values in Table 2 so negative? This makes this reviewer wonder about the noise in the assay.

Minor issues

1.       The grammatical issues are better but remain.   For example, Line 87 “Ana o 1 has a low cashew content”  was probably meant to say “Cashew has a low Ana o 1 content”.

2.       Simply showing the sequences of Ana o 2 and Ana o 3 does not appear to be useful.  Table 1 is sufficient.

3.       Do not see the value of Figure 5.

4.       Figure 3B is not a typical CD figure.  Since this appears to be a CD analysis of a combination of at least 2 proteins (Figure 2), it is hard to see the relevance.